# Sport in the Laryngectomized Patient: A Literature Review and Single Case Presentation

**DOI:** 10.3390/jpm13060982

**Published:** 2023-06-12

**Authors:** Massimo Mesolella, Salvatore Allosso, Grazia Salerno, Gaetano Motta

**Affiliations:** 1Unit of Otorhinolaryngology, Department of Neuroscience, Reproductive Sciences and Dentistry, University Federico II of Naples, 80131 Naples, Italy; grace@unina.it; 2Unit of Otorhinolaryngology, Department of Mental and Physical Health and Preventive Medicine, University Luigi Vanvitelli, 80131 Naples, Italy; gaetano.motta@unicampania.it

**Keywords:** total laryngectomy, sport, rehabilitation, laryngeal cancer, swimming

## Abstract

Background: Total laryngectomy is an operation that involves numerous problems for the patient, especially in daily life: loss of the fact, loss of voice, evident scars and persistence of the tracheostoma. Much is known about rehabilitation programs involving the voice, swallowing, shoulder girdle rehabilitation; less explored is the field of sport and sports rehabilitation in the laryngectomized patient. Methods: We conduced systematic review was conducted according to the Preferred Reporting Items for Systematic Reviews and Meta-Analyses (PRISMA) statement in order to evaluate the possibilities of practicing sports for the patient after total laryngectomy. Results: From an initial search of 4191 papers, we have come to include six papers for this literature review. We have also reported one of our clinical cases referring to a laryngectomized patient who swims competitively at an amateur level even after surgery with a particular device. The purpose of this work is to understand the role and importance of sport in rehabilitation and the possibilities that a frail patient like the laryngectomized patient has in practicing sport. Surely the best results are obtained in subjects who practiced sports before surgery. Conclusion: It is evident that sport is important in the psychological and motor recovery of the laryngectomized patient. There is still a lack of clear rehabilitation protocols, especially for water sports, which allow all laryngectomized patients to return to sports. We believe that early resumption of physical activity makes the experience of the disease less dramatic.

## 1. Introduction

Laryngeal cancer accounts for approximately 30% of the malignant neoplasms in the head and neck region and 2% of all malignancies. Also it causes 83,000 deaths per year worldwide. Estimates of around 5000 new cases of laryngeal cancer are expected in Italia in 2022–2023 period. Considering the incidence and mortality rates described in the world, they are variable and detected by exposure to alcohol and tobacco. For this reason, the mortality from cancer of the larynx in males varies 10–15 times in male patients among the various European countries [1].

In the Developed World, 50% to 60% of patients present with early laryngeal cancer, defined by the American Joint Committee on Cancer as a T1 or T2 tumor without nodal involvement or distant metastases [2].

The impact of advanced laryngeal cancer and its extensive surgical treatments cause significant morbidity for these patients. Although the percentage of patients who undergo total laryngectomy has decreased over the years in favor of conservative surgery, total laryngectomy remains a devastating event due to the impact on daily life [3,4,5,6]. Total laryngectomy is a radical method reserved for the treatment of advanced laryngeal carcinoma or as a salvage therapy.

Once the skin of the neck has been incised and the selective neck dissection has been performed, we arrive at the laryngeal structures. The hyoid bone is identified cranially, preserving the insertion of the sternohyoid muscle. The io-thyroid-epiglottic space and the thyroid-hyoid membrane are dissected in order to access the glosso-epiglottic vallecula. Below, the surgical limit reaches up to the first tracheal ring. Posteriorly, the cricoid is identified and the overlying mucosa is sectioned, separating the larynx from the esophagus. The trachea is permanently attached to the skin. The continuity of the digestive tract is restored by packing the pharyngostoma [7].

This type of surgery leads to significant anatomic and functional alterations, such as permanent tracheostoma, voice, senses and feeding problems. Additionally, this surgery brings about aesthetic and psychological consequences, and patient perception of physical health is significantly damaged. The permanent division between the airway and the digestive tract, with the presence of the permanent tracheostomy, determines a series of problems for patients such as: loss of voice, difficulty in swallowing, greater risk of respiratory tract infection, loss of smell and reduction of taste, the ability to perform a Valsalva maneuver, change in the perception of oneself and alteration of social life [8,9,10].

According to Jayasuriya et al. (2010), a total laryngectomy is described as a stressful experience, resulting in the complete loss of natural voice and a subsequently poor quality of life, that changes the patient’s social roles forever. Laryngectomies are performed in order to lengthen a patient’s lifetime but at the expense of committing to a completely different lifestyle than before. Length of survival is no longer a sufficient measurement. The expected and potential future quality of life must also be evaluated and given significant weight [11].

Also, important aspects of life, such as social relations, communication, sexuality and emotional status are also negatively affected. Hence, the total laryngectomy has a major impact on the patient’s quality of life (QoL). QoL has a particular relevance for those diagnosed and treated for advanced laryngeal cancer because of the myriad difficulties these individuals experience with everyday functioning. However, multiple factors beyond breathing, speech and swallowing affect QOL in individuals with advanced laryngeal cancer, including time since diagnosis, treatment type, methods of coping, mental health, social support, gender and cultural issues [12,13,14].

In the study conducted by Mallis A., the various factors influencing the long-term life of 92 laryngectomees patients were considered. He reported that the most common complaint in the functional disorders was olfactory disorders (69.6%) followed by taste disorders (34.8%) and throat dryness (34.8%). Swallowing and eating disorders were also reported by a significant percentage of patients—15.2% and 13% respectively—while pain was rarely reported as a problem (4.2%). Various communication difficulties were also reported: most of the patients reported a difficulty in communicating with strangers (56.5%) or over the telephone (78.3%); only a lower percentage (30.4%) experienced communication difficulties, even in the closest family nucleus. Family issues only rarely reported by their patients (10.9%), although 23.9% reported adverse effects in sex lives. About the social interactions, only the 23.9% of patients confessed to decreased participation in social events although a minority (2.2%) reported problems with their friends. Regarding the psychological status, the 58.7% of patients reported uneasiness due to their appearance while 23.9% felt embarrassed due to their voice or disease, while 30.4% mentioned a feeling of loneliness. A percentage of 67.4 reported that their mood worsened after surgery in contrast to 32.6% of the patients who reported no change (28.3%) or a better mood (4.3%). A percentual of 80.5% of the patients reported worsened financial situation, with the majority (91.3%) of the patients also reporting decreased capacity for work in accord with the high retirement rate (65.2%). Patients’ general sense of life changes in 93.5% of the patients: Their life was changed to a greater (65.2%) or lesser degree (28.3%) as opposed to 6.5% who reported no perceived changes in their lifestyle. Considering in general the perception of the quality of life of the patients, it changed in 93.5% of the subjects: 65.2% of the patients reported a change to a greater extent in their life; 23.8% reported a minor change; only 6.5% of patients did not detect any change in their lifestyle [15,16].

One of the important aspects to consider in the laryngectomy patient is the possibility of practicing sports after total laryngectomy surgery [17,18].

From the data in the literature, it is evident that not much attention is paid to the patient’s sports rehabilitation. It is true that, on average, the patient with laryngeal carcinoma in most cases does not follow a healthy lifestyle or engage in sports activities; however, it is true that in the literature there are a few cases described of people who have restarted playing sports after total laryngectomy [14,17,18,19,20].

Restriction of participation in activities imposed by laryngectomy may negatively impact quality of life. One group of persons negatively affected by laryngectomy is avid water enthusiasts. Specifically, Buntzel et al. [19] reported 19 (29%) of 65 laryngectomy patients surveyed were avid swimmers before surgery. Of these 19 patients, 53% indicated that the inability to swim after the laryngectomy led to a decreased sense of “life satisfaction”.

It is known from the literature that sport is a very important factor in the prevention, management and prognosis of cancer [21]. Indeed, it is evident that physical exercise should be integrated into the therapeutic programs of all those living with and beyond cancer [15]. The first approach is to understand what are the physical and emotional effects that cancer has had on the individual. From this awareness arises the need for a new culture for both the patient and the doctor that it is necessary to undertake a rehabilitation path to sport under the guidance of qualified personnel [22].

This study aims to revise the literature to identify the possibilities that the laryngectomy patient has to practice sports and what types of sports are practiced to date. Furthermore, the aim is also to formulate indications to improve the quality of life of the laryngectomized patient.

## 2. Materials and Methods

This systematic review was conducted according to the Preferred Reporting Items for Systematic Reviews and Meta-Analyses (PRISMA) statement [23]. The study was carried out in accordance with the principles of the Helsinki Declaration, and an informed consent was obtained from our participant before including him in the study. No review protocol was registered for this study.

Manuscripts were screened primarily by Ovid Medline and EMBASE and from other sources (PubMed central, Web of Science, Cochrane review and Google Scholar) and published from January 1980 to July 2022. Literature searches were performed in August 2022.

The search was conducted in PubMed with the last search taking place on 26 July. The keywords searched for on PubMed are: “sport after total laryngectomy” AND “Laryngectomy” OR “Total Laringectomy” AND “sport” OR “Gym”. To minimize the risk of missing relevant data, a cross-reference search of the selected articles was performed, and the “cited by” function on Google Scholar was also used to obtain other relevant articles for the study.

Works for which neither the abstract nor the complete text were available were excluded. An initial screening of titles and abstracts and a subsequent full-text screening were both done by two independent assessors. The inclusion criteria were primary search studies (including descriptive study, observational study, case studies, books) published after January 1980. We excluded secondary research studies (e.g., review articles, consensus conferences, lecture, letter). Only articles with full text available and in English language were included. We excluded all the articles that did not meet the inclusion criteria or deal directly with the issue investigated. All disagreements were resolved by discussion.

Two independent reviewers (MM ad SA) conducted the electronic search. All articles were initially screened by title and abstract. Then, studies that were believed to be relevant to our search were downloaded and the full-text manuscripts reviewed to determine eligibility. The conflict between reviewers was resolved by a third author (GS). Data extraction from the included studies was systematically made using a structured form by two independent reviewers (MM and SA). If data were missing from the articles, then the corresponding author was contacted in an attempt to obtain the data. The following data were extracted, when available: author and year of publication, country, number of patients, gender (male or female), age (years), sport practiced, period, observation time, kind of device, reported accident.

## 3. Results

### 3.1. Systematic Review Study Selection

Our research after duplicate removal focused on 1881 studies. We excluded 203 articles due to time of publication and then 1619 were finally screened. The subsequent exclusion of articles due to the above criteria led to the definition of six articles. We summarized the included studies in Figure 1.

Roger F. Gray et al. describes one 40-year-old man, an avid swimmer. To allow humans to return to swimming, a special device was built from a diving tube. The device was provided with a watertight endotracheal portion and a valve that prevented the entry of water. This hose is held with a mouthpiece between the lips. The patient became able in water training in 8 weeks. Swimming in the laryngectomy patient is possible, but it is right to practice it under the expert guidance of trained rescue personnel. However, this first article in the literature has encouraged the knowledge and production of swimming devices in laryngectomized patients [24].

Amir M. Karamzadeh describes the experience of four people who practice water sports after total laryngectomy. All four patients practiced water sports after laryngectomy surgery. Two use a special device (Larkel Breathing Devices), and one uses a handmade device. Two patients used their fingers to occlude the stoma. Only one patient reported an accident while playing sports. The study shows how the passion for swimming preceding laryngectomy is a very important factor in returning to the water. One of the patients also learned to stay on the water without occluding the stoma by coordinating breathing. The patient describes her abilities as “a necessity for survival” [14].

Basile N. Landis et al. have published a case of a 68-year-old man with a passion for swimming and motorcycling who, a few months after his laryngectomy, built two DIY devices for practicing sports. The first when using scuba diving equipment is a flexible tube that goes from the trachea to the mouth, allowing him to breathe through his nose. The second is a tea strainer that fixed to the tracheostoma prevents the entry of insects. Thanks to the two devices the patient is able to practice swimming and snorkeling and continues to ride a motorcycle [18].

Crevenna et al. tried to establish rules for introducing hydrotherapy in the rehabilitation of the laryngectomized patient. Six male patients aged between 47 and 75 were selected who practiced group hydrotherapy with musical accompaniment under the guidance of a physiotherapist. The sessions were held three times a week for an observation period of 8 weeks. All patients used Larkel Breathing Devices. The program aimed to improve endurance capacity; postural control; flexibility; mobility; symptoms such as daytime fatigue/exhaustion/tiredness; expectoration; and quality of life. Although the study demonstrates that hydrotherapy is an excellent choice for the rehabilitation of the laryngectomized patient, today it is not a practice used. The aim of the study was to evaluate the feasibility of group hydrotherapy in laryngectomized patients in compliance with safety and acceptance [20].

W. Hagner et al. selected 30 patients aged between 48 and 74 in a rehabilitation program lasting 14 days that included both specific rehabilitation exercises for the shoulder, neck, pectoral muscles, as well as language rehabilitation and sports rehabilitation with volleyball, cycling, tennis, archery, football, badminton, ringo and an obligatory evening walk. Using the Harvard Step Test, which is recognized as the best index for determining fitness, they achieved statistically significant improvements over 14 days of therapy. The result was better in the younger age group. These data demonstrate how the type of rehabilitation protocol used is valid for rehabilitating the laryngectomized patient [25].

J. Buntzel et al. enrolled 38 patients, 5 women and 33 men aged between 41 and 79 years. Of these, 31/38 already practiced sports as children, and 34/38 had learned to swim at a young age. Sixteen of them had used our offered hydrotherapy program. 10 patients reported about walking and cycling with their families or friends. Swimming (12 patients), cycling (5 patients) and walking (20 patients). Of the 16 patients who participated in the hydrotherapy program, 15 obtained favorable responses. Of these, 4 started swimming. All patients were provided with swimming devices. The results obtained by the 15 patients: reduction of shoulder and neck pain after 3–4 hydrotherapy sessions. 14/16 patients recommended an aquatic experience as post-surgical rehabilitation [19]. According to the authors, it is evident that sport helps in the rehabilitation process of the cancer patient [12,26]. The authors highlight that cycling and walking are important new disciplines in the rehabilitation of the laryngectomized patient. Both sports offer the improvement of the shoulder and spine muscles and can be practiced without the help of expert figures. Bunzel et al. show how sports rehabilitation in the laryngetomized patient is the first step towards a normal life, and hydrotherapy is a novelty in the rehabilitation treatment in patients with laryngeal cancer [19].

From the studies analyzed, three proposed sports therapy as rehabilitation in the laryngectomized patient [19,20,25]; the others described experiences of subjects returning to sports after laryngectomy [14,18,24]. It is evident that the return to sports is conditioned by the patient’s passion for it before surgery [14,18,19,20,24,25].

Table 1 provides a résumé for the articles included in the qualitative analysis and summarizes the most relevant results.

### 3.2. Istitutional Case Report

DG aged 60, underwent total laryngectomy and bilateral lateral cervical emptying for adenoid cystic carcinoma which had subglottic localization (Figure 1). The patient was an assiduous swimmer, enrolled in an amateur sports club, had achieved good results in numerous category competitions Seniors. After the operation, the patient experienced an initial period of depression despite the psychological support of our health facility and family members. After having overcome the distrust towards the resumption of sporting activity due to fears about its feasibility, he accepted the invitation to resume the sporting activity. A special device was therefore used (Figure 2). It establishes a watertight seal with a double-cuffed rigid tracheostomy tube. The configuration is similar to a snorkel device. The cuffed tracheostomy tube connects to a flexible hose with an attached mouthpiece. The mouthpiece is gripped with the teeth and sealed by the lips. Air flows through the nose, around the palate, into the mouth, through the Larkel tubing and into the lungs.

This device is designed for use in pool aquatic activities and is not recommended for activities such as ocean swimming. We prefer the flexible tube with a rigid walled cuffed tracheostomy tube. The rigid device allowed overinflation of the cuff to help immobilize the tube within the trachea, decreasing the likelihood of tube dislodgment and at the same time maintaining a patent lumen. The head and neck straps help to keep the device in proper position while swimming. The tracheal tube and cuff permitted a watertight seal while at the same time providing a conforming and comfortable fit.

Currently, the patient continues to swim at an amateur competitive level (Figure 3).

## 4. Discussion

The total laryngectomy was going to change the patient’s life forever. Often the patient was not educated enough on the forefront of the surgery to understand the impacts a laryngectomy would have on his life.

While improved cure rates, prolonged disease-free survival and organ preservation are the primary focus of the treatment of advanced laryngeal cancer, the implicit purpose of organ preservation is improved function and QoL. We must also consider that the conservation of the organ does not always correspond to the conservation of its function. In fact, preserving the laryngeal organ does not always correspond to preserving the ability to swallow, phonate and breathe, consequently an improvement in the quality of life is not implied [8]. For this reason, a complete understanding and assessment of the patient is necessary, taking into account all these considerations in order to facilitate treatment, assistance, rehabilitation and eventual end-of-life care [27].

The social implications of laryngectomy are many; it negatively influences communication in various social contexts, such as work and community life. These relational difficulties are often responsible for determining a progressive isolation of the individual. A laryngectomee patient may need to interrupt meals several times both with family and friends. Furthermore, the presence of a tracheostoma, in addition to determining an increase in work disabilities, can alter the ability to carry out common activities of daily life such as skiing, gardening, singing in a choir [28,29].

The effects of interpersonal relationships and socialization activities in determining an improvement in the quality of life in laryngectomees patients are known. Furthermore, the involvement of the partner or family members can increase the patient’s success in rehabilitation, and strangers to the patient can have a negative impact regarding the patient’s emotional involvement and communication skills [5,28,30]. Helvik, changes in one’s physical appearance may result in alterations in body-image and self-concept, which may affect one’s socialization and QoL.

Total laryngectomy leads to decreased physical activity of patients. Generally, rehabilitation programs are fundamentally based on the improvement of voice with tracheoesophageal prosthesis, electrolarynx or esophageal voice and swallowing.

One of the major problems is also that the laryngectomized patient would seem to be precluded from participating in sports, especially aquatic ones. The most striking experience is definitely swimming. The presence of the tracheostoma represents a challenge for the doctor and the patient in the management of sports practice, especially for swimming [28].

As early as 1981 Nigel Edward described two types of devices for laryngectomized subjects: the extension tube that protrudes above the water (like a “snorkel tube”); a short extension tube that must be held directly between the teeth and allows the physiological breathing path to be restored [17].

Despite the presence of studies on the description of swimming devices and their reliability, there are still few data on the number of patients who are rehabilitated to practice sports after laryngectomy [17,31].

## 5. Conclusions

The introduction of sport in rehabilitation programs in the laryngectomized patient is certainly important in improving the quality of life. Certainly, hydrotherapy, especially if practiced in a group and under the guidance of expert personnel, can be practiced by the laryngectomized patient prepared for this experience. Introducing sports rehabilitation, together with psychological support for the patient and family-caregivers, can have a positive impact on the rehabilitation of the laryngectomized patient.

The possibility of returning to sport after an operation, such as total laryngectomy, is certainly a good starting point for the patient in order to overcome the dramatic event of the disease, even psychologically.

The possibility of practicing sport gradually but at intervals established in the rehabilitation protocols can lead to a lasting improvement in the QoL of the cancer patient. Being able to practice group sports, even at an amateur competitive level, helps the patient recover their skills and above all gives comfort in the daily struggle that the cancer patient faces. The idea that, despite the disease, it is possible to lead a life more and more like normal is an encouragement for those who experience the same dramatic event.

The scientific research of new materials and devices specific for each patient and suitable for each need is of fundamental importance in making disability less limiting.

## Data Availability

Data sharing not applicable. No new data were created or analyzed in this study. Data sharing is not applicable to this article.

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
