# Peer review of "Sport in the Laryngectomized Patient: A Literature Review and Single Case Presentation"

_jpm, 2023, doi:10.3390/jpm13060982_

Round 1

Reviewer 1 Report

Dear Authors,

Sport in general is a very powerful therapeutic for maintaining mental and physical health. For laryngectomized people, and especially for those for whom it was an integral part of life before the operation, returning to sports is not only important in physical rehabilitation but also in maintaining satisfaction. Therefore, I support this and similar analyzes and believe that the work should be accepted. During the review, I did not find any significant complaints.

Author Response

Thank you for the work done on the review.

Reviewer 2 Report

Thank you for the opportunity to review this paper. My major concern is that the authors state they are completing a systematic literature review, but in truth, it seems that this is more of an integrative literature review and case study. I suggest revising both the title and abstract to reflect this change. For example, the tile could be “Sport in the laryngectomized patient: A literature review and single case presentation.”

Abstract

Can the authors add more information into the study design regarding their single case study. It is not mentioned at all in the abstract. The results would also benefit from something about what kind of studies were found (many single case designs) which then supports the design of this paper.

Introduction

Lines 27-30- please add citations for all of the stats listed.

Line 40- can you describe a laryngectomy and what structures are removed? This will help the reader to understand the physical impact of the procedure.

Line 99- purpose statement could use revision. I would clarify that the purpose is to also make recommendations to improve QOL of laryngectomized patients.

Materials and Methods

Why was 1980 chosen as a start date for this search?

If the authors would like to continue to call this a systematic review, please provide search string information for each database as an appendix.

If this is a systematic review (instead of just a literature review) I have concerns that the search terms are not comprehensive enough to include all forms of exercise. For example, walking and running are exercises that might not be considered “sport” but still may be relevant to the goal of this work. I have concerns that these types of articles may be missed with your search strings.

If the authors would like to continue to call this a systematic review, they would need to perform a bias assessment of the studies in the review.

Results-

I suggest reviewing the literature FIRST (section 3.2) and then going into your case study (section 3.1) as this provides a foundation for what you are also presenting.

Section 3.2, is initially missing a summary of what kinds of studies, the years, number of subjects, etc., were found.

Please add a sub-heading called “Study Summary Information” to your section beginning on line 170, as this section consists of summaries of each of the studies. Considering re-working this section to group similar studies together into single paragraphs that describe similar sports.

Discussion

The first four paragraphs of the discussion is just information that should be in (or is already in) the introduction. Lines 301-327. Suggest removing this, or integrating it into the introduction.

Further integration of results is necessary. Please also identify next steps needed to ensure maintained QOL in laryngectomized patients.

English quality is excellent.

Author Response

REVIEWER 2

Thank you for the opportunity to review this paper. My major concern is that the authors state they are completing a systematic literature review, but in truth, it seems that this is more of an integrative literature review and case study. I suggest revising both the title and abstract to reflect this change. For example, the tile could be “Sport in the laryngectomized patient: A literature review and single case presentation.”

  • I have revised the title and corrected as suggested

Abstract

Can the authors add more information into the study design regarding their single case study. It is not mentioned at all in the abstract. The results would also benefit from something about what kind of studies were found (many single case designs) which then supports the design of this paper.

  • I have correct (Line 20)

Introduction

Lines 27-30- please add citations for all of the stats listed.

  • correct

Line 40- can you describe a laryngectomy and what structures are removed? This will help the reader to understand the physical impact of the procedure.

  • LINE 48: Once the skin of the neck has been incised and the selective neck dissection has been performed, we arrive at the laryngeal structures. The hyoid bone is identified cranially, preserving the insertion of the sternohyoid muscle. The io-thyroid-epiglottic space and the thyroid-hyoid membrane are dissected in order to access the glosso-epiglottic valleculae. Below, the surgical limit reaches up to the first tracheal ring. Posteriorly, the cricoid is identified and the overlying mucosa is sectioned, separating the larynx from the esophagus. The trachea is permanently attached to the skin. The continuity of the digestive tract is restored by packing the pharyngostoma. 5

Line 99- purpose statement could use revision. I would clarify that the purpose is to also make recommendations to improve QOL of laryngectomized patients.

  • CORRECT line 114

Materials and Methods

Why was 1980 chosen as a start date for this search?

If the authors would like to continue to call this a systematic review, please provide search string information for each database as an appendix.

If this is a systematic review (instead of just a literature review) I have concerns that the search terms are not comprehensive enough to include all forms of exercise. For example, walking and running are exercises that might not be considered “sport” but still may be relevant to the goal of this work. I have concerns that these types of articles may be missed with your search strings.

If the authors would like to continue to call this a systematic review, they would need to perform a bias assessment of the studies in the review.

  • The date 1980 was chosen because the first article that talks about sport in the laryngectomized patient dates back to 1982. There are no other articles prior to this date. We believe it is useless to add research so as not to add anything new.

It is not possible to consider walking a sport. For this reason we have explained what the search criteria were. LINE 128 The keywords searched for on PubMed are: “sport after total laryngectomy” AND “Laryngectomy” OR “Total Laringectomy” AND “sport” OR “Gym”. To minimize the risk of missing relevant data, a cross-reference search of the selected articles was performed, and the “cited by” function on Google Scholar was also usd to obtain other relevant articles for the study.

Results-

I suggest reviewing the literature FIRST (section 3.2) and then going into your case study (section 3.1) as this provides a foundation for what you are also presenting.

  • done

Section 3.2, is initially missing a summary of what kinds of studies, the years, number of subjects, etc., were found.

Please add a sub-heading called “Study Summary Information” to your section beginning on line 170, as this section consists of summaries of each of the studies. Considering re-working this section to group similar studies together into single paragraphs that describe similar sports.

  • We have not thought of further summarizing the studies because a summary table is already present which distributes the patients according to the year of publication of the article. For this reason, the criterion for enumeration of the articles is not based on the similar sport but chronological. therefore we would like to leave this part as it is.

Discussion

The first four paragraphs of the discussion is just information that should be in (or is already in) the introduction. Lines 301-327. Suggest removing this, or integrating it into the introduction.

  • correct

Further integration of results is necessary. Please also identify next steps needed to ensure maintained QOL in laryngectomized patients.

  • correct

Reviewer 3 Report

Radical surgery of the larynx leaves important consequences for the quality of life of the patient. This permanent tracheostomy requires special care and an adapted way of life. The manuscript reveals both the impediments of an incomplete postoperative rehabilitation as well as the importance of resuming physical activities at the end of the treatment.

Author Response

Thanks for the review and your comment

Round 2

Reviewer 2 Report

Thank you for your attention to all of my comments and concerns.

However, I still do not believe this review meets the PRISMA criteria of "systematic review" reporting. Please see check list for prisma here: https://prisma.shinyapps.io/checklist/  Specifically, there study lacks report of full search strings, and a bias assessment. With no bias assessment, there is no critical review of the quality of the studies. A systematic literature search (what the authors did to search the databases) is just one part of several kinds of literature reviews, including systematic reviews, scoping reviews, literature reviews, etc., and in itself, does not make a study a full "systematic review." I once again, suggest the authors temper their language surrounding the use of "systematic review" as they performed a scoping review/literature review of the topic area. Suggest change of the title as well as comments saying they followed PRISMA. Consider following this scoping review reporting checklist instead:

Tricco, A.C.; Lillie, E.; Zarin, W.; O’Brien, K.K.; Colquhoun, H.; Levac, D.; Moher, D.; Peters, M.D.J.; Horsley, T.; Straus, S.E.; et al. PRISMA Extension for Scoping Reviews (PRISMA-ScR): Checklist and Explanation. Ann. Intern. Med. 2018, 169, 467–473.

English is proficient.

Author Response

Dear Reviewer,
Thank you for your time.
I agreed to also review the title according to your indications and you can verify it in the new uploaded work.
however I cannot find corrections for some of your objections.
1. I used the prisma checklist for the research and drafting of the work, as reported in my references.
2. it is true that no biases are analysed. I believe that it is not possible to analyze any bias given that in the literature there are only 6 works found that speak of sport, of the most disparate types of sport in laryngectomised patients. The real bias is unique: perhaps the argument in the literature is not well documented. That is, there are many more cases of people who practice sports that are not described, nor are there rehabilitation protocols in water sports in the laryngectomised patient. In this mare magnum of absence of literature I would like to place my study to encourage discussion and description.
Thank you. I hope it's understandable on your part to be able to take the job.
